# Real-Time Pipeline Fault Detection in Water Distribution Networks Using You Only Look Once v8

**DOI:** 10.3390/s24216982

**Published:** 2024-10-30

**Authors:** Goodnews Michael, Essa Q. Shahra, Shadi Basurra, Wenyan Wu, Waheb A. Jabbar

**Affiliations:** Faculty of Computing, Engineering and Built Environment, Birmingham City University, Birmingham B4 7RQ, UK; goodnews.michael@mail.bcu.ac.uk (G.M.); shadi.basurra@bcu.ac.uk (S.B.); wenyan.wu@bcu.ac.uk (W.W.)

**Keywords:** object detection, YOLOv8, image analysis, CNN, annotation, water management system

## Abstract

Detecting faulty pipelines in water management systems is crucial for ensuring a reliable supply of clean water. Traditional inspection methods are often time-consuming, costly, and prone to errors. This study introduces an AI-based model utilizing images to detect pipeline defects, focusing on leaks, cracks, and corrosion. The YOLOv8 model is employed for object detection due to its exceptional performance in detecting objects, segmentation, pose estimation, tracking, and classification. By training on a large dataset of labeled images, the model effectively learns to identify visual patterns associated with pipeline faults. Experiments conducted on a real-world dataset demonstrate that the AI-based model significantly outperforms traditional methods in detection accuracy. The model also exhibits robustness to various environmental conditions such as lighting changes, camera angles, and occlusions, ensuring reliable performance in diverse scenarios. The efficient processing time of the model enables real-time fault detection in large-scale water distribution networks implementing this AI-based model offers numerous advantages for water management systems. It reduces dependence on manual inspections, thereby saving costs and enhancing operational efficiency. Additionally, the model facilitates proactive maintenance through the early detection of faults, preventing water loss, contamination, and infrastructure damage. The results from the three conducted experiments indicate that the model from Experiment 1 achieves a commendable mAP50 of 90% in detecting faulty pipes, with an overall mAP50 of 74.7%. In contrast, the model from Experiment 3 exhibits superior overall performance, achieving a mAP50 of 76.1%. This research presents a promising approach to improving the reliability and sustainability of water management systems through AI-based fault detection using image analysis.

## 1. Introduction

Efficient water management systems are vital for the effective distribution and conservation of water resources [1]. However, the detection and repair of faulty pipelines present significant challenges for water utilities, leading to water losses, infrastructure damage, and increased operational costs [2]. Traditional manual inspection methods are often time-consuming and insufficient [3]. Advancements in artificial intelligence (AI) and computer vision offer promising solutions to automate pipeline inspections [4]. This study introduces an AI-based model that utilizes image analysis to detect faults in water management pipelines [5]. By employing computer vision techniques, deep learning algorithms, and image processing, the model aims to automate fault detection, facilitating proactive maintenance and reducing water losses [6,7]. Integrating AI into water management systems offers numerous benefits, including improved efficiency, cost savings, enhanced accuracy, and timely response to infrastructure issues [8,9]. Early detection of pipeline faults is crucial due to the significant water wastage and resource strain caused by leaks, cracks, and other defects [10]. Timely identification of these issues allows water utilities to minimize water losses, conserve resources, and mitigate risks such as water contamination and property damage [11]. The AI-based model analyzes images to identify specific fault patterns, making it a valuable tool for addressing critical water management challenges [12].

The research builds on relevant studies that have advanced the understanding and application of AI-based models for pipeline fault detection. For example, authors in [13] proposed an AI-based water leak detection system utilizing cloud information management to enhance leak detection efficiency and accuracy. Similarly, authors in [14] focused on predicting pipeline failures in water supply networks using logistic regression and support vector classification. The AI-based model for detecting pipeline faults through images involves several stages [15]: data collection and integration, data pre-processing and augmentation, utilization of YOLOv8 for object detection [16], transfer learning techniques, dataset annotation and labeling, model training and validation, and deployment and integration into existing water management systems. Each component contributes to the accuracy, reliability, and efficiency of the fault detection process [17,18]. While implementing an AI-based model for pipeline inspection presents challenges such as data availability, class imbalance, generalization, and real-time inference, the benefits and potential applications outweigh these difficulties [19]. Early fault detection enables proactive maintenance, reducing water losses, infrastructure damage, and associated costs. Integrating the AI model with pipeline monitoring systems facilitates remote inspection, efficient allocation of maintenance resources, and real-time decision-making [20].

This paper aims to explore case studies and real-world implementations of AI-based pipeline inspection systems, demonstrating their effectiveness in detecting pipeline faults, streamlining maintenance efforts, and optimizing water management operations. Understanding the capabilities and implications of AI-based models for pipeline fault detection will help water utilities and stakeholders make informed decisions about adopting these technologies, ultimately enhancing the efficiency, sustainability, and resilience of their water management systems. The structure of this paper is as follows: Section 2 presents the most recent related work, Section 3 explains the methodology applied, Section 4 explains the experimental design, Section 5 elaborates on the results and discussion from all AI models, and Section 6 presents a detailed discussion of the results, including comparisons with findings from other studies. Finally, Section 7 concludes the work and outlines the new directions for future research.

## 2. Related Works

This review highlights the applications of AI techniques in water resource management, emphasizing their potential to improve practices by detecting pipeline faults, optimizing water distribution, and enhancing efficiency and sustainability. Vanijjirattikhan et al. [21] proposed an AI-based water leak detection system utilizing cloud information management to enhance leak detection efficiency and accuracy. The system trains machine learning algorithms on leakage sound data, with the Deep Neural Network (DNN) outperforming the Support Vector Machine (SVM) and matching the performance of the Convolutional Neural Network (CNN) with a simpler structure. Field trials showed that novice operators achieved over 90% leak pinpointing accuracy, comparable to experts. Hu et al. [22] introduced the DBSCAN-MFCN method for detecting leakages in urban water supply networks. This approach combined density-based spatial clustering (DBSCAN) and multiscale fully convolutional networks (MFCN) to reduce complexity by zoning the water network. The DBSCAN-MFCN method outperformed traditional techniques like SVM, NBC, and KNN, improving detection efficiency by 78%, 72%, and 28%, respectively. Wang et al. [23] developed a framework for assessing sewer conditions using computer vision and deep learning on CCTV inspection videos. The framework, employing Faster R-CNN and semantic segmentation models, achieved high accuracy in defect detection with average precision, recall, and F1-scores of 88.99%, 87.96%, and 88.21%, respectively. This system supports automated sewer assessment and maintenance planning, consistent with professional inspector evaluations. Niu and Feng [24] evaluated five AI methods (ANN, ANFIS, ELM, GPR, and SVM) for daily streamflow prediction. Their study showed that SVM, GPR, and ELM outperformed ANN and ANFIS, highlighting the importance of selecting appropriate models based on reservoir characteristics. Ahn et al. [25] discussed high-frequency Acoustic Emission (AE) systems for pipeline condition monitoring, combining Genetic Algorithm (GA) for feature selection, Principal Component Analysis (PCA) for preprocessing, and AI and SVM for fault classification. They introduced Intensified Envelope Analysis (IEA) as a signal-preprocessing method, showing that GA and IEA improved classification performance over PCA and envelope analysis.

Robles-Velasco et al. [26] proposed incorporating failure probability and consequences into a comprehensive tool for optimal pipe replacement planning. Applied to a Spanish city’s water supply network, this methodology suggested that replacing 3% of the network’s pipes annually could prevent about 30% of failures, demonstrating a feasible approach to reduce unexpected failures. Shukla and Piratla (2020) addressed the challenge of leak detection in buried pipelines using a deep-learning algorithm. Their approach involved scalogram images of vibration signals from accelerometers on the pipeline surface, achieving up to 95% detection accuracy for PVC pipelines. This method minimized human intervention in leak detection. All papers reviewed above have been summarized in Table 1.

## 3. Methodology

The process of object detection involves several critical stages, as illustrated in Figure 1. It begins with data collection, where a comprehensive dataset of labeled images is curated, ensuring that it includes diverse and representative samples of the objects to be detected. In the data preprocessing stage, images are resized, normalized, and augmented through techniques such as flipping, rotation, and scaling to improve the model’s robustness and generalization capabilities. During model selection, a suitable architecture, such as YOLOv8, is chosen based on the specific task requirements, carefully balancing speed and accuracy. The training and validation phase involves dividing the dataset into training and validation sets, training the model on the former, and tuning hyperparameters while monitoring performance on the latter to prevent overfitting. Finally, the evaluation stage uses metrics like mean average precision (mAP), precision, recall, and F1-score to rigorously assess the model’s performance, ensuring it meets the necessary standards for accuracy and reliability in detecting objects in new, unseen images.

### 3.1. Data Collection

In this work, high-resolution images were captured using drones to encompass diverse pipeline locations and scenarios, providing the raw data for the YOLOv8 model. The dataset is meticulously organized into three directories: ‘train’, ‘val’, and ‘test’. The training set comprises 248 images, the validation set includes 60 images, and the test set contains 38 images, all uniformly resized to dimensions of 640 × 640 pixels, shown in Figure 2. Initially, the dataset is unlabeled and requires thorough annotation. For the purpose of this work, two object categories are defined: ‘pipe’ and ‘faulty pipe’, ensuring precise classification and detection of pipeline conditions.

### 3.2. Data Annotation

Data annotation is vital for developing object detection models, as it involves labeling objects within images to provide the model with the necessary information for learning. Object detection, a fundamental task in computer vision, involves identifying and locating objects within images or videos. YOLOv8 is known for its speed and accuracy in these tasks. Creating high-quality annotations is crucial as the dataset quality significantly impacts the model’s performance.

#### 3.2.1. Annotation Stages

The annotation process includes the following steps:Selection of Annotation Tool: An appropriate tool is chosen to facilitate the labeling process. Common tools include LabelImg, VGG Image Annotator (VIA), and Labelbox.Labeling Objects: Annotators mark and label objects of interest within the images. For pipeline inspection, this may include pipes, valves, junctions, or anomalies.Annotation Format: Annotations follow specific formats such as Pascal VOC, COCO, or YOLO. In this project, the YOLO format is used, including class labels, bounding box coordinates, and image dimensions.Quality Control: A quality control process ensures accuracy and consistency through regular checks and reviews of annotated images. Annotators use guidelines and reference images to maintain consistency.

#### 3.2.2. Annotation Process for the Data Directory

Upon completing the annotation stages, careful tool selection is crucial to avoid errors during model execution. For this project, the LabelImg tool was initially employed. The LabelImg library provides a user-friendly interface with primary functions accessible on the left side of the window.

The annotation operation sequence: “Open Dir” > “Create RectBox” > “Enter Category Name” > “Save” as presented in Figure 3. During the annotation process for images in the “train” and “valid” directories using the LabelImg tool, several issues were encountered. The user interface occasionally glitched, requiring reloading and causing an incomplete label directory. To mitigate this, annotations were performed in batches and consolidated into a single directory. However, this led to inconsistencies in label categories, with some batches having two categories and others only one. As the model expected two label categories, these inconsistencies triggered errors, necessitating a complete redo of the annotation process, which could take weeks depending on the dataset’s size. To address these challenges, the CVAT annotation tool was used. CVAT offers similar annotation processes and formats to LabelImg but allows for pausing and resuming annotations, and labels are predefined at the outset to ensure consistency. After completing the annotations, the label directory is exported in YOLO format, ensuring a streamlined and consistent process as shown in Figure 4.

### 3.3. Dataset Splitting

The annotation of the dataset provides information about the objects present in each image. These typically include the following:Bounding Boxes: Annotations specify the coordinates of bounding boxes around the objects of interest in the images. Bounding boxes are represented by pairs of coordinates, often in the format (x_min, y_min, x_max, y_max), which define the top-left and bottom-right corners of the box.Class Labels: Each object within a bounding box is associated with a class label. In this case, ‘pipe’ which is represented as ‘0’, and faulty_pipe which is denoted as ‘1’. Class labels help the model distinguish between different object categories.Image Information: Annotations often include information about the image itself, such as the image dimensions or a unique image identifier.

The text file containing bounding box coordinates and object classes is associated with a distinct image identifier. The labels for the training and validation datasets are placed in separate directories named ‘train’ and ‘valid’, respectively. These directories are located within another directory labeled ‘labels’. Simultaneously, the image directories for training, validation, and testing are stored in a directory called ‘images’. It is crucial to adhere to this precise folder structure and nomenclature for the YOLOv8 model to successfully locate and utilize the dataset. Subsequently, we generate a YAML file that includes paths to the directories housing the training, validation, and testing images, along with the associated object classes. This ensemble of images, labels, and the YAML file is organized within a dedicated ‘YOLO’ directory. This annotated dataset serves as the foundation for training the YOLOv8 model, enabling it to acquire the capability to detect and categorize objects within the images.

### 3.4. Data Visualization

Data visualization is essential in the scientific method, as effective visual representations help scientists understand their data thoroughly and communicate their findings clearly [28]. The image aspect ratio, which is the ratio of an image’s width to its height, is a key attribute in fields such as photography, design, computer vision, and web development. This ratio affects how an image is displayed across different screens and media. Ensuring uniform image sizes is critical for training the YOLOv8 model. Figure 5 confirms that all images in the training and validation directories have an aspect ratio of 1.0, indicating uniformity in image dimensions without any significantly large or small sizes. It is essential to visualize the distribution of bounding box aspect ratios in both the training and validation label directories, as this provides insights into the range of box sizes. Any irregularities in the annotations may necessitate a re-evaluation of the annotation process. Figure 6 indicates that most bounding boxes correspond to relatively smaller objects, indicating that the process can move forward as planned.

### 3.5. YoloV8 Architecture

The architecture of YOLOv8, as shown in Figure 7, builds on the foundation of YOLOv5 with key enhancements, particularly the introduction of the C2f module, which combines high-level features with contextual information to boost detection accuracy [29]. YOLOv8 utilizes an anchor-free model, directly predicting object centers with a decoupled head that processes objectness, classification, and regression tasks independently, leading to improved overall accuracy [30]. The output layer uses a sigmoid function for objectness scores and a softmax function for class probabilities. For loss functions, YOLOv8 employs CIoU and DFL for bounding box loss and binary cross-entropy for classification loss, yielding better performance, especially for detecting smaller objects. Additionally, YOLOv8 introduces YOLOv8-Seg, a semantic segmentation model using the CSPDarknet53 feature extractor and C2f module, achieving state-of-the-art results in both object detection and semantic segmentation while maintaining high speed and efficiency [31]. YOLOv8 can be executed via the command line interface (CLI) or as a PIP package, offering integrations for labeling, training, and deployment. In terms of performance, YOLOv8x achieved an average precision (AP) of 53.9% on the MS COCO test-dev 2017 dataset with a 640-pixel image size, outperforming YOLOv5, which had an AP of 50.7% on the same input size. This evaluation, conducted on an NVIDIA A100 with TensorRT, demonstrated a processing speed of 280 frames per second (FPS) [32].

## 4. Experimental Design

We begin by installing Ultralytics to initiate the setup of the YOLOv8 model after preparing our data for training. Ultralytics is a popular open-source framework for working with YOLOv8 and other YOLO variants. It simplifies the implementation and training of object detection models, offering features like multi-scale training, multi-GPU support, and integration with PyTorch. The YOLO models can be customized to suit specific tasks, adjusting architecture, hyperparameters, and data augmentation. YOLOv8 introduced five different scaled variants, which are YOLOv8n (nano), YOLOv8s (small), YOLOv8m (medium), YOLOv8l (large), and YOLOv8x (extra-large).

YOLOv8m was chosen as it provides a balanced trade-off between detection accuracy and computational efficiency, making it ideal for real-time applications like pipeline fault detection. It offers improved accuracy compared to smaller models while maintaining faster inference times than larger models, ensuring timely detection without overburdening computational resources.

For this training, the YOLOv8m (medium) model is loaded from a pre-trained checkpoint file named “yolov8m.pt” and assigned to the variable “model”. The training process is initiated by calling the variable model and setting the parameters of the base model. The output of the training process is stored in the result variable.

### 4.1. Experiment for Base Model

In the context of real-time object detection, certain parameters need to be considered carefully to optimize performance. Key parameters include input size, which determines the dimensions of images processed by the model; smaller sizes generally result in faster inference times due to the reduced number of pixels. Batch size is another critical factor, as it refers to the number of images processed simultaneously during inference. A smaller batch size can lead to reduced latency, allowing for quicker responses essential in real-time applications.

Additionally, the model size—whether small, medium, or large—affects both speed and accuracy. Smaller models are typically faster and more memory-efficient, making them suitable for real-time applications, particularly on devices with limited resources. The confidence threshold is also vital, as it sets the minimum score for predictions to be deemed valid. Lowering this threshold can increase the number of detections, but it may also result in more false positives. Striking the right balance is crucial to ensuring that only relevant and confident detections trigger responses, thus enhancing overall efficiency.

In the context of real-time detection and proactive maintenance, the confidence threshold plays a critical role in balancing early fault detection and system reliability. A lower threshold can help detect potential faults earlier by identifying issues even when the confidence level is low, making it useful for catching early-stage problems.

Finally, the mean average precision (mAP) provides insight into the model’s overall performance in detecting objects across various classes and Intersection over Union (IoU) thresholds. While mAP does not directly impact real-time processing speed, a high score reflects a robust model capable of effective detection in real-time scenarios. By meticulously evaluating these parameters, developers can significantly enhance the effectiveness and reliability of real-time object detection systems. Table 2 shows the other important parameters used.

### 4.2. Training Visualization

The visualization process in YOLOv8 training provides valuable insights into the model’s performance and learning progress. By tracking metrics like precision, recall, and mAP@50, and visualizing predictions with confidence scores, one can effectively monitor and improve the model’s object detection capabilities. This process ensures that the model is accurately and reliably detecting objects, which is crucial for practical applications such as pipeline inspection using drone-captured images. Figure 8, illustrates the initial labeling of the dataset prior to the commencement of the training process. These labels represent the ground truth, serving as the benchmark for evaluating the model’s performance. As shown in Figure 9, the progression of the YOLOv8 model’s learning during training is depicted, highlighting the evolution of its predictive accuracy over the course of several iterations. In the early stages of training, the model’s predictions exhibit lower accuracy, as the model is still in the process of learning to identify and differentiate between the various features and object classes in the dataset. However, as training progresses, there is a noticeable improvement in prediction accuracy, which is reflected in the increasing percentage of correctly identified objects. This improvement signifies the model’s growing ability to generalize from the labeled data, effectively capturing the underlying patterns associated with pipeline defects.

## 5. Results

### 5.1. Results from Experiment 1

The results include a weight folder containing the optimal model to be used for prediction stored in a folder-“Best.pt”. The fitness score is determined by a weighted blend of four metrics, which include [Precision (P), Recall (R), mAP at 0.5 IoU threshold (mAP@0.5), and mAP from 0.5 to 0.95 IoU thresholds (mAP@0.5:0.95)] as shown in Table 3. The confusion matrix, normalized confusion matrix, F1 confidence curve, precision curve, recall curve, precision-recall curve, and loss functions have been measured and evaluated as shown in Figure 10. The training time for this experiment was 4.773 h.

In Figure 11, the X-axis represents the confidence threshold values from 0 to 1, and the Y-axis represents the corresponding F1-scores. We can see that as we adjust the confidence threshold, the F1-score changes, reflecting the trade-off between precision and recall. Initially, at low thresholds, the F1-score is low due to high false positives. As the confidence threshold increases, the F1-score improves, eventually reaching an optimal peak where the balance between precision and recall is achieved. This optimal balance occurs for all classes, resulting in an F1-score of approximately 60 at a confidence threshold of 0.316.

In Figure 12, the X-axis represents the confidence threshold values from 0 to 1, and the Y-axis represents the corresponding precision. As the confidence threshold increases, the precision improves. At low thresholds, many predictions are classified as positive, including many false positives, resulting in lower precision. As the threshold increases, the model becomes more conservative, reducing false positives and improving precision. This conservative trend is observed across all classes, yielding a precision of 1.0 at a confidence threshold of 0.753. Further increasing the confidence threshold does not significantly enhance precision, as the model is already highly selective at this point.

In Figure 13, the X-axis represents the confidence threshold values from 0 to 1, and the Y-axis represents the corresponding recall. At low thresholds, recall is high because the model classifies almost all predictions as positive, including many true positives. As the threshold increases, recall decreases because the model becomes more selective, missing some true positives.

Figure 14, the X-axis represents recall, and the Y-axis represents precision. At the beginning of the curve (lower recall values), precision is typically high because the model is highly confident in its few positive predictions. As recall increases (moving right on the X-axis), the model identifies more true positives but may also include more false positives, causing precision to decrease. The curve often shows a trade-off between precision and recall: increasing recall reduces precision and vice versa.

Based on Figure 15, the evaluation of the Experiment 1 model demonstrated significant effectiveness in the domains of object detection and classification, with notable performance improvements across all metrics. During the training process, we observed a consistent decline in both localization and classification losses, indicating that the model was effectively learning and refining its predictions. This trend continued into the validation phase, where the model exhibited strong generalization capabilities on previously unseen data. Both precision and recall metrics showed steady increases, signifying that the model not only identified more relevant objects over time but also did so with increasing accuracy. These improvements were further validated by rising average precision scores across various IoU thresholds, from moderate to strict, underscoring the model’s robustness. The comprehensive advancements across all metrics highlight YOLOv8’s efficiency in handling complex object detection tasks. The results suggest that this model is highly applicable to real-world scenarios, delivering reliable and accurate detections across diverse conditions.

### 5.2. Results from Experiment 2

In this training process, we load the YOLOv8l (large) model from a pre-trained checkpoint file labeled “yolov8l.pt” and associate it with the variable “model1”. We made the following adjustments to the model’s training configuration: the batch size was reduced from 16 to 8, and the number of training epochs was extended to 100 to enhance optimization. The AP score for each label and the mAP at iou 50 and iou 75 threshold with a training time of 17.621 h, as shown in Table 4.

### 5.3. Results from Experiment 3

To train this model, the batch size was set to 64 and epochs at 50 as a decline in performance was seen after the iou threshold of 50 for Experiment 2. The iou threshold parameter was set to 0.2 rather than 0.7. The Ap score for each label and the mAP at iou 50 and iou 75 threshold with a training time of 4.947 h, as shown in Table 5.

## 6. Discussion

Three models were trained with different parameter configurations to enhance their accuracy.

Experiment 1: Serving as the base model, it was trained for 50 epochs, with a batch size of 16, and took 4.773 h. The mean average precision (mAP) at an intersection over union (IoU) threshold of 50% for the “pipe” label, “faulty pipe” label, and both labels combined were 65.1%, 90.0%, and 74.7%, respectively.Experiment 2: This experiment involved training for 100 epochs, using a batch size of 8, and took 17.621 h. The mAP at a 50% IoU (Intersection over Union) threshold for the “pipe” label, “faulty pipe” label, and both labels combined were 63.6%, 80.1%, and 72.4%, respectively.Experiment 3: In this experiment, the model was trained for 50 epochs with a batch size of 64, and the training duration was 4.947 h. The mAP at a 50% IoU threshold for the “pipe” label, “faulty pipe” label, and both labels combined were 69.1%, 83.2%, and 76.1%, respectively.

The experiments offered valuable insights into how various training parameters impacted model performance, especially in object detection accuracy. Adjustments in batch size and epochs had a marked influence on outcomes. Smaller batch sizes allowed more frequent weight updates, enabling the model to capture finer patterns, but introduced instability during extended training. Conversely, larger batch sizes provided more stable learning but sometimes missed subtle details. Similarly, increasing the number of epochs enhanced accuracy but also increased the risk of overfitting if not carefully managed. Key configuration elements like learning rate and data augmentation techniques played a crucial role in determining the model’s adaptability and effectiveness across different scenarios. Fine-tuning these parameters was essential for achieving consistent and reliable results, emphasizing the importance of a balanced approach in the experimental design. Regarding performance evaluation, a higher IoU threshold (above 50%) resulted in decreased detection accuracy, particularly for the “pipe” label, where predicted bounding boxes often fell short of the 0.7 IoU threshold used during training. Nevertheless, Table 6 shows that the model in Experiment 1 achieved a strong mAP50 of 90% for detecting faulty pipes, though its overall mAP50 was 74.7%. In comparison, the model from Experiment 3 exhibited better overall performance with a mAP50 of 76.1%, making it a leading candidate for further testing and possible deployment. These findings underscore the critical role of parameter optimization in enhancing detection capabilities within water management systems.

### 6.1. Testing the Model

Experiments on an AI-based faulty pipe detection system using the YOLOv8 model revealed a clear distinction between faulty and normal pipes. The model achieved a robust detection rate of 90% for faulty pipes as shown in Figure 16, demonstrating its effectiveness in identifying critical anomalies crucial for maintenance and safety applications. However, the model’s accuracy in detecting normal pipes varied significantly, ranging from 51% to 75% as shown in Figure 17a and Figure 17b, respectively. This inconsistency suggests potential challenges in consistently recognizing normal pipes, likely due to insufficient representation in the training dataset or visual overlap with non-target elements.

These findings emphasize the need for further refinement in model training and parameter tuning to enhance its performance, particularly in achieving more consistent detection across different conditions. While the model shows great potential for critical fault detection, there is also a clear opportunity to improve its general detection accuracy to ensure reliable performance in a broader range of scenarios.

### 6.2. Comparison with Others

We discussed the performance of YOLOv8 in comparison to other models as shown in Table 7. YOLOv8 has emerged as a standout model in the realm of object detection, particularly for pipeline fault detection in water management systems. With an impressive speed of 280 frames per second (FPS), YOLOv8 is exceptionally fast, making it highly suitable for applications that require real-time processing, such as surveillance and autonomous navigation. The model’s accuracy is noteworthy, achieving a mean average precision (mAP) ranging from 72% to 76.1%. This performance places YOLOv8 on par with contemporary detection models, addressing the critical need for precision in various object detection tasks. Additionally, YOLOv8’s lightweight architecture facilitates efficient deployment, especially in environments with limited computational resources, making it a versatile choice for practical applications. Its high degree of optimization for real-time detection further enhances its effectiveness in scenarios where rapid decision-making is crucial, demonstrating its strengths in speed, accuracy, and robustness.

In contrast, R-CNN operates at a significantly slower speed of 2–3 FPS, rendering it unsuitable for real-time applications. While it compensates for this limitation with respectable accuracy, achieving an mAP of 66–69%, R-CNN is better suited for tasks that prioritize object localization quality over speed. The larger model size presents challenges for deployment in resource-constrained environments, necessitating more powerful computational capabilities. Moreover, the high complexity of R-CNN models demands substantial computational resources, limiting their application to environments equipped with the necessary infrastructure. As a result, R-CNN excels in scenarios where precision is paramount, particularly in high-quality object localization tasks.

Faster R-CNN offers a modest improvement over its predecessor, achieving speeds of 7–9 FPS. Although this model can handle slightly more complex tasks, it still falls short of real-time capabilities. The accuracy of Faster R-CNN ranges from 70% to 75% mAP, making it suitable for general object detection applications. However, like R-CNN, it also suffers from a larger model size, which can affect deployment in resource-limited settings. The moderate speed allows it to be viable for applications that can tolerate slight delays, yet its medium complexity makes it more accessible than R-CNN while still requiring significant computational power. Consequently, Faster R-CNN serves effectively in general object detection tasks across various domains, providing a balance between accuracy and speed.

SSD (Single Shot Detector) operates at a speed of 22–30 FPS, allowing it to achieve real-time detection capabilities. However, it does not match the performance of YOLOv8 in terms of speed and accuracy. With an accuracy range of 40–50% mAP, SSD may not provide sufficient precision for high-stakes applications that require high detection rates. The medium size of the SSD model allows for a balance between resource requirements and detection capabilities, facilitating deployment in various environments. Its medium complexity also enables easier implementation, making it accessible for a wide range of applications. SSD is particularly suitable for scenarios where a balance between speed and accuracy is needed, such as in mobile applications that may not demand the highest levels of precision.

In summary, the comparison of these object detection models highlights the distinct advantages and limitations each one presents. YOLOv8 stands out for its speed and accuracy, making it a prime candidate for real-time object detection tasks. Meanwhile, R-CNN and Faster R-CNN, while offering solid performance in specific applications, are hindered by their slower processing speeds and higher resource demands. SSD offers a balance between speed and accuracy but does not reach the performance levels of YOLOv8. The findings underscore the importance of selecting the appropriate model based on specific use-case requirements, taking into account factors such as speed, accuracy, and computational complexity to meet the diverse needs of various object detection applications.

## 7. Conclusions and Future Work

YOLOv8 demonstrated strong potential in detecting pipeline faults within water management systems, consistently delivering robust performance across various experimental settings. Notably, Experiment 3 highlighted the model’s adaptability under different configurations, achieving the highest average performance with an mAP50 of 76.1%. This success was attributed to the strategic tuning of batch size and epoch duration, effectively balancing accuracy and efficiency. The findings emphasize the model’s strengths, including its speed, real-time detection capabilities, and resilience under diverse conditions such as varying lighting and camera angles. Meticulous dataset preparation and annotation were critical in enhancing the model’s accuracy, especially in distinguishing between normal and faulty pipes. Combined with YOLOv8’s advanced features, such as its anchor-free architecture, these efforts led to high detection rates, particularly in identifying critical anomalies essential for proactive maintenance. While the model shows promise, challenges remain as it moves toward real-world deployment. Ensuring its scalability for larger operations, addressing class imbalance, and minimizing noise in data are key areas for future work.

Research will focus on optimizing the model for broader scenarios, refining data augmentation techniques, and exploring more sophisticated hyperparameter tuning. Collaboration with industry stakeholders will be essential to fine-tune the model’s real-time processing and integrate it seamlessly into existing water management infrastructures. Additionally, limitations identified in this study, such as inconsistencies in detecting normal pipes, present opportunities for further improvement. Addressing these issues will enhance the model’s reliability and efficiency, paving the way for its application in water management systems worldwide.

## Figures and Tables

**Figure 1 sensors-24-06982-f001:**
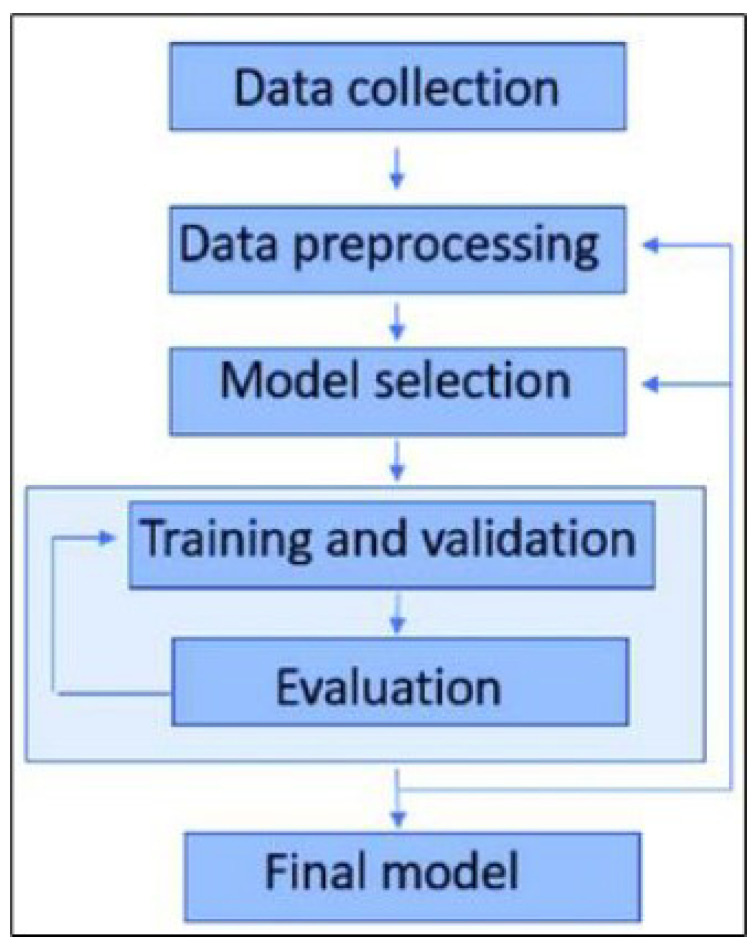
Traditional pipeline for object detection (Yolov8).

**Figure 2 sensors-24-06982-f002:**
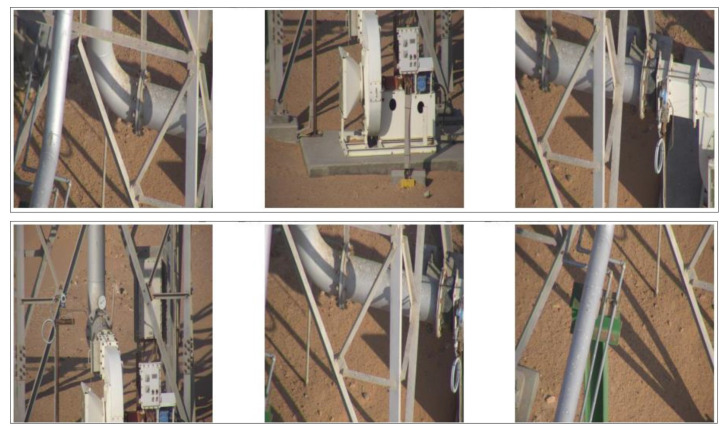
Samples from dataset.

**Figure 3 sensors-24-06982-f003:**
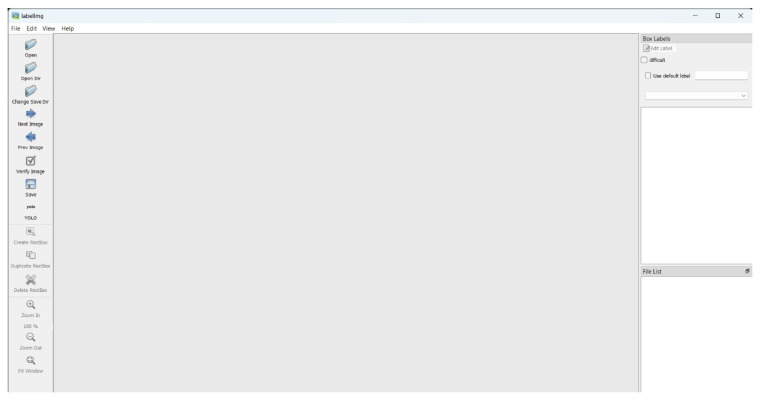
Label image UI.

**Figure 4 sensors-24-06982-f004:**
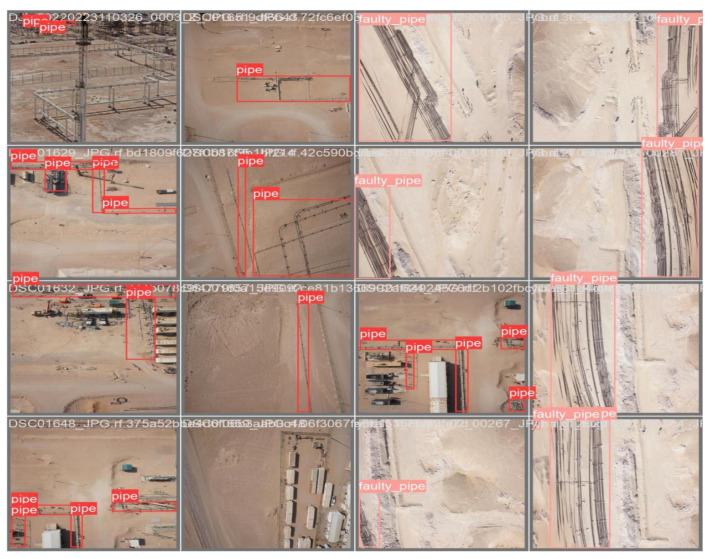
Labeled data after annotation.

**Figure 5 sensors-24-06982-f005:**
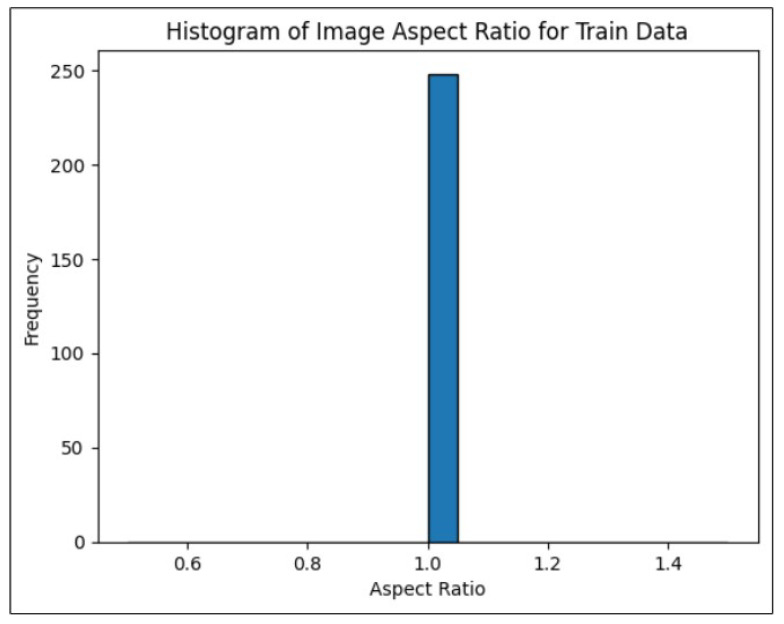
Histogram of image aspect ratio for validation data.

**Figure 6 sensors-24-06982-f006:**
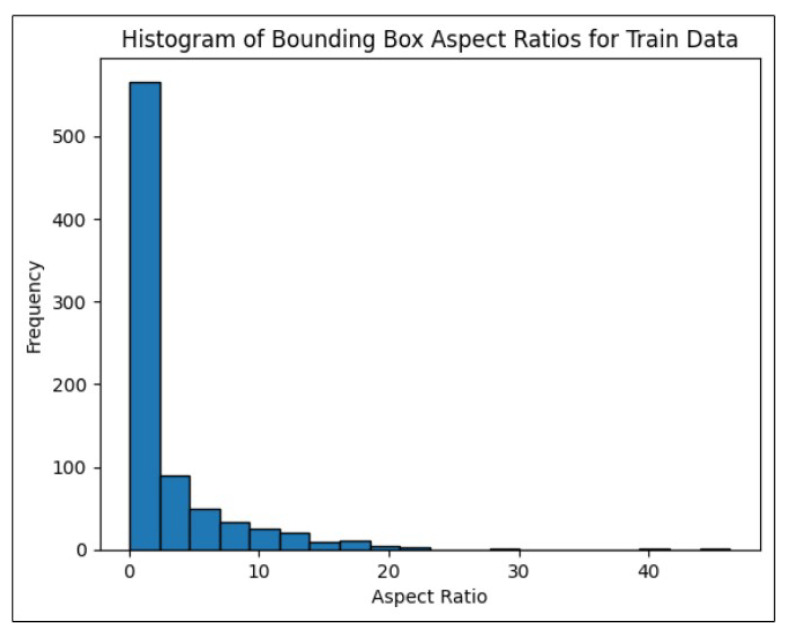
Histogram of bounding box aspect ratio for training data.

**Figure 7 sensors-24-06982-f007:**
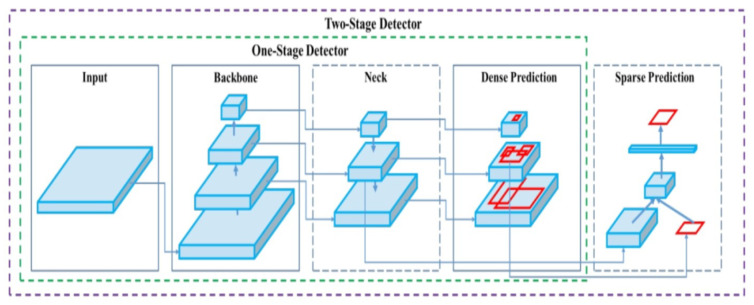
The architecture of Yolov8.

**Figure 8 sensors-24-06982-f008:**
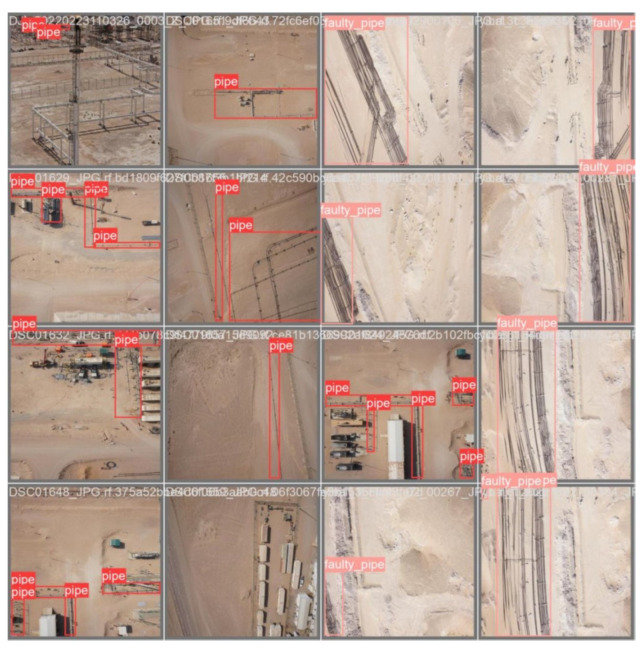
Label data before training.

**Figure 9 sensors-24-06982-f009:**
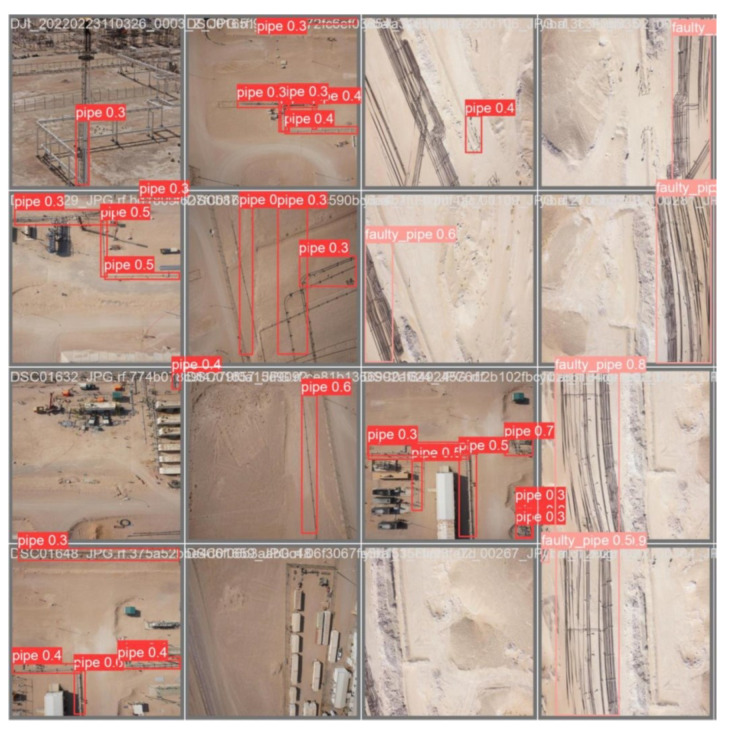
Predicted label during the training Batch 0.

**Figure 10 sensors-24-06982-f010:**
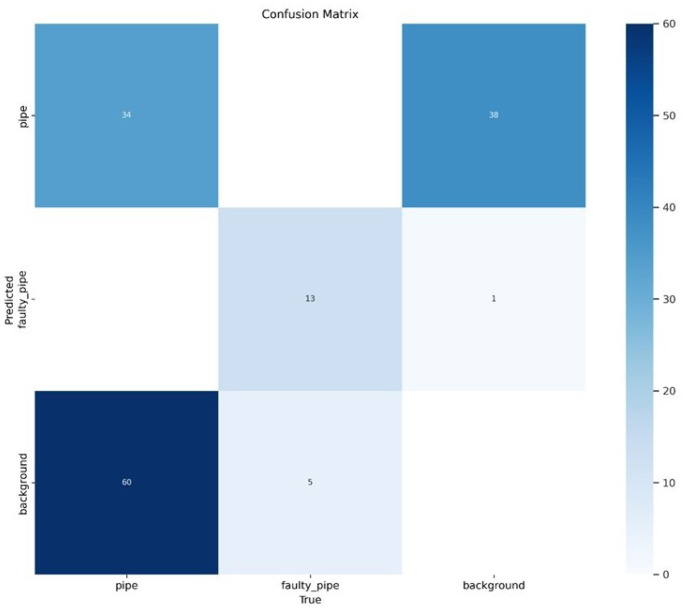
Confusion matrix.

**Figure 11 sensors-24-06982-f011:**
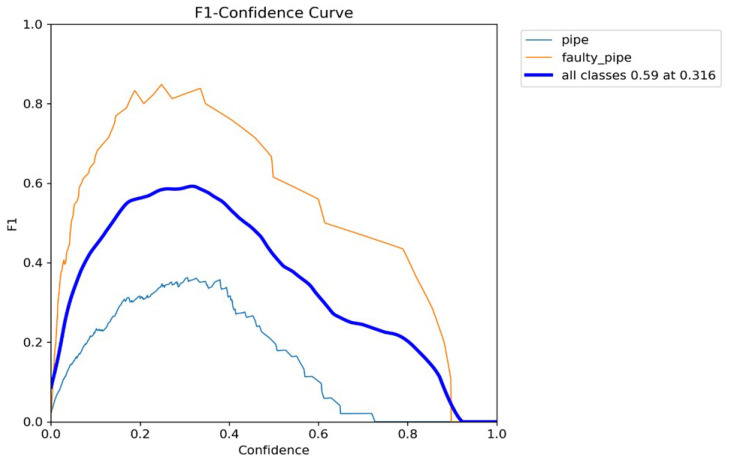
F1 confidence curve.

**Figure 12 sensors-24-06982-f012:**
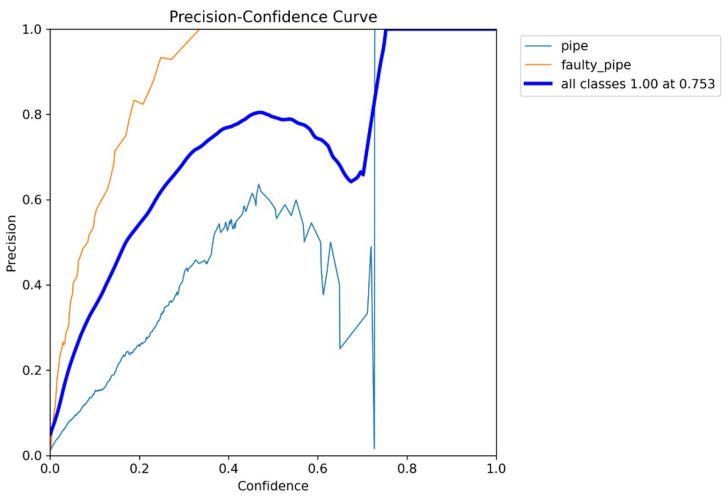
Precision confidence curve.

**Figure 13 sensors-24-06982-f013:**
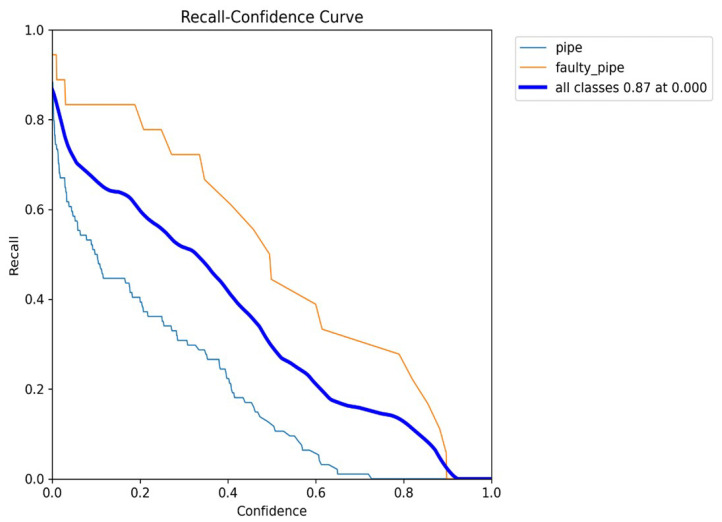
Recall confidence curve.

**Figure 14 sensors-24-06982-f014:**
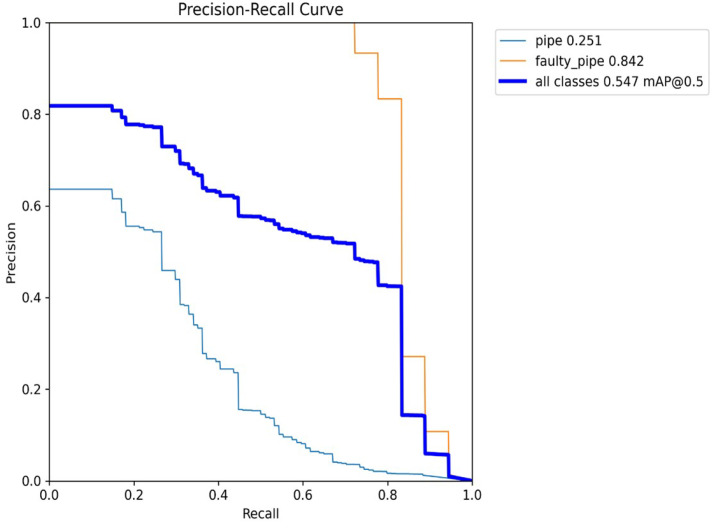
Precision vs. recall curve.

**Figure 15 sensors-24-06982-f015:**
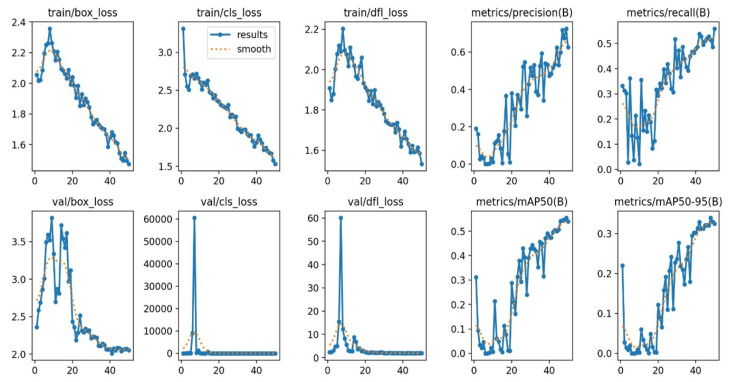
Loss function vs. mAP.

**Figure 16 sensors-24-06982-f016:**
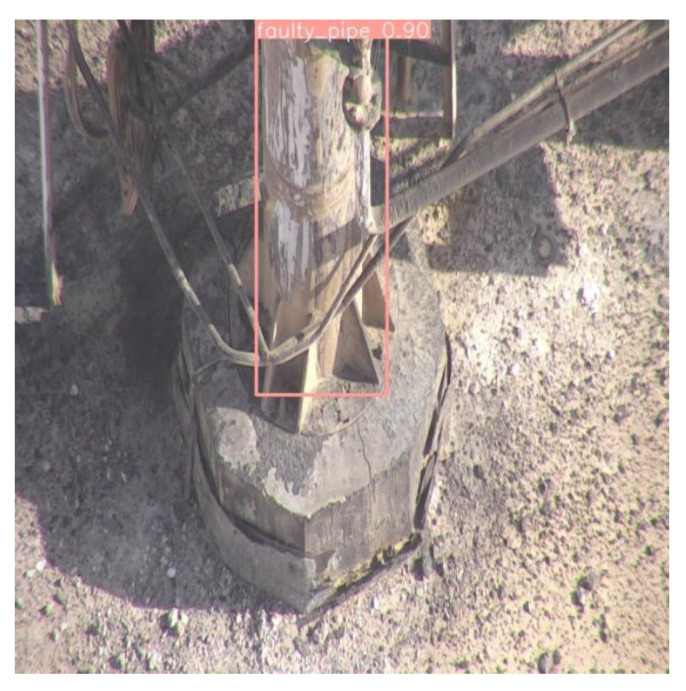
Test image detecting a faulty pipe.

**Figure 17 sensors-24-06982-f017:**
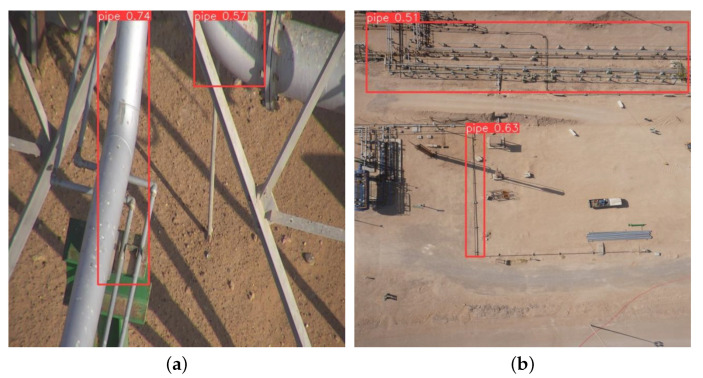
Test image detecting a pipe.

**Table 1 sensors-24-06982-t001:** Summary of research papers used in the literature review.

Ref.	Topic	Implemented AI	Results
[21]	AI-based acoustic leak detection in water distribution systems	Deep learning algorithms and CNNs	Effectiveness of deep learning techniques in accurately detecting pipeline faults
[23]	Towards an automated condition assessment framework of underground sewer pipes based on closed-circuit television (CCTV) images	R-CNN (Regional Convolution Neural Network)	Average precision, recall, and F1-score of 88.99%, 87.96%, and 88.21%, respectively
[22]	Novel leakage detection and water loss management of urban water supply network using multiscale neural networks	DBSCAN-MFCN, which combines DBSCAN algorithm with MFCN algorithm; SVM, Naïve bayes Classifier (NBC), k-Nearest Neighbor (KNN)	DBSCAN-MFCN accuracy is 78% over SVM, 72% over NBC, 28% over KNN
[24]	Evaluating the performances of several AI methods in forecasting daily streamflow time series for sustainable water resource management	AANN, ANFIS, Extreme Learning, ELM, GPR, SVM	All five AI methods achieved satisfactory forecasting results; however, SVM, GPR, and ELM outperform ANN and ANFIS in terms of the chosen evaluation benchmarks
[25]	AI-based machine learning considering flow and temperature of the pipeline for leak early detection using acoustic emission	GA, PCA SVM	The performance of GA for feature selection and IEA preprocessing is superior to PCA and envelope analysis in terms of fault classification accuracy
[27]	Leakage detection in water pipelines using supervised classification of acceleration signals	CNN model adapted from pre-trained Alexnet network	CNN model achieves 95% accuracy in detecting PVC leaks and 98% accuracy using carefully selected acceleration signal data
[26]	Prediction of pipe failures in water supply networks using logistic regression and support vector classification	Logistic regression and support vector classification	SVM achieved the highest WQC prediction accuracy at 97.01%

**Table 2 sensors-24-06982-t002:** Experiment setup for the model training.

Parameter	Value
task:	detect
mode:	train
model:	yolov8m.pt
epochs:	50
patience:	50
batch:	16
imgsz:	640
optimizer:	auto
verbose:	true
close_mosaic:	10
fraction:	1.0
overlap_mask:	true
mask_ratio:	4
iou:	0.7
max_det:	300
augment:	false
boxes:	true
box:	7.5
format:	torchscript
momentum:	0.937
weight_decay:	0.0005
warmup_epochs:	3.0
warmup_momentum:	0.8
mosaic:	1.0

**Table 3 sensors-24-06982-t003:** Summary of Experiment 1.

Class	Images	Instances	P	R	mAP50
all	48	112	0.76	0.71	0.74
pipe	48	94	0.71	0.68	0.65
faulty pipe	48	18	0.85	0.82	0.90

**Table 4 sensors-24-06982-t004:** AP and mAP for Experiment 2.

Class	Images	Instances	P	R	mAP50
all	48	112	0.78	0.74	0.72
pipe	48	94	0.70	0.66	0.63
faulty pipe	48	18	0.77	0.0.75	0.80

**Table 5 sensors-24-06982-t005:** AP and mAP for Experiment 3.

Class	Images	Instances	P	R	mAP50
all	48	112	0.76	0.68	0.76
pipe	48	94	0.70	0.62	0.69
faulty pipe	48	18	0.88	0.72	0.83

**Table 6 sensors-24-06982-t006:** Results comparison.

	Training Time/h	%mAP50 for Pipe	%mAP50 for Faulty Pipe	%mAp50 for All
Experiment 1	4.773	65.1	90.0	74.7
Experiment 2	17.621	63.6	80.1	72.4
Experiment 3	4.947	69.1	83.2	76.1

**Table 7 sensors-24-06982-t007:** Comparsion with other works.

Ref.	Model	Speed (FPS)	Accuracy (mAP %)	Model Size	Real-Time Suitability	Complexity
Our study	YOLOv8	280	72–76	Light- weight	Yes	Low
[33]	R-CNN	2–3	66–69	Large	Not	High
[34]	Faster R-CNN	7–9	70–75	Large	Moderate speed	Medium
[35]	SSD Single Shot	22–30	40–50	Medium	Yes	Medium

## Data Availability

The data presented in this study are available on request from thecorresponding author.

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
