# Peer review of "Real-Time Pipeline Fault Detection in Water Distribution Networks Using You Only Look Once v8"

_sensors, 2024, doi:10.3390/s24216982_

Round 1
Reviewer 1 Report
Comments and Suggestions for Authors
This study introduces a YOLOv8 model for detecting pipeline defects, specifically targeting leaks, cracks, and corrosion through image analysis. Detecting pipe failures in water distribution networks is a critical research area, as fault detection is essential for ensuring the reliable supply of clean water. Therefore, the research has practical application value. However, the paper has several issues that should be addressed:
1.
The paper situates its research question within the context of existing studies but does not present a groundbreaking or radically novel hypothesis. While it employs deep learning techniques that are widely used in image processing, the methodology lacks innovative modifications to the YOLO model. Instead, it applies established methods to address a specific problem. From a theoretical perspective, it does not propose new hypotheses or theories that significantly advance the understanding of deep learning or image processing.
2.
The title, "Real-Time Pipeline Fault Detection in Water Distribution Networks Using YOLOv8," suggests a focus on real-time detection, yet the paper does not adequately discuss real-time aspects of fault detection.
3.
The abstract mentions proactive maintenance through early fault detection to prevent water loss, contamination, and infrastructure damage. However, the paper itself lacks sufficient discussion on early fault detection.
4.
The experiments do not include comparisons with other models, which weakens the evaluation of the proposed approach.
Author Response
Please kindly see the attached file. It includes the responces for all comments.

Reviewer 2 Report
Comments and Suggestions for Authors
Traditional pipeline detection methods are time-consuming, costly, and prone to errors. The development of artificial intelligence and computer vision provides a solution for automated pipeline detection. This paper elaborates in detail on how to use drones and YOLO V8 for pipeline detection from the aspects of data collection, data annotation, dataset splitting, and YOLO v8 architecture. This work is meaningful and the reviewer recommends publication after the authors address the following comments/questions:
l The author attempts to address practical engineering issues by applying YOLOV8; however, from the perspective of this article, there is no clear indication of its scientific contribution. It is suggested that the author delve deeper to reflect the scientific significance of this research in the text.
l The use of drones to collect image data in the article is an innovative approach, but there is a lack of detailed description regarding the specific environment and conditions of data collection, such as the geographical location of the pipelines, the surrounding environment, and the influence of light. These factors may impact the quality of the data and the performance of the model.
l During the data annotation process, although the encountered problems and solutions are mentioned, it could be further elaborated on how to ensure the accuracy and consistency of the annotation, as well as the measures taken to verify the quality of the annotation.
l From the perspective of image recognition, the dataset used in the article is relatively small, and the images are only obtained through drones in specific scenarios, which may result in insufficient diversity of the data and difficulty in covering various pipeline fault situations in practical scenarios. The author is requested to comment on the generalization performance of the proposed method.
l The article conducts three experiments to train the model. The reviewer believes that the experimental setup is relatively simple, and the dimensions of comparison are not rich enough. Moreover, the explanation for the selection and adjustment of experimental parameters is insufficient. For example, a more detailed explanation should be provided regarding the basis and expected effects of reducing the batch size from 16 to 8 and extending the epochs to 100 in Experiment 2.
Comments on the Quality of English LanguageThe quality of English language is acceptable.
Author Response

(The authors gave the same response as above.)

Reviewer 3 Report
Comments and Suggestions for Authors
Major Revision
This manuscript presents an interesting real-time fault detection model based on the YOLOv8 network architecture. While the manuscript addresses an interesting and valuable research topic, it requires substantial revisions to strengthen the technical details, contextual framing, and rigor of the analysis and discussion.
(1) The rationale for selecting the YOLOv8m version among the five variants mentioned in the experimental design should be clarified, and the limitations of the study need to be more rigorously defined and justified.
(2) For abstract, specific advantages listed in lines 12-13 should be integrated into the body of the text to provide a more concise and focused summary.
(3) The introduction could benefit from a broader and more comprehensive discussion of the research problem, supported by citations to YOLO application references (Mask YOLOv7-Based Drone Vision System for Automated Cattle Detection and Counting; Artificial Intelligence and Applications. Transforming unmanned pineapple picking with spatio-temporal convolutional neural networks; Computers and Electronics in Agriculture. ).
(4) The writing can be simplified by reducing compound sentences and nested clauses to improve
(5) For computer vision detectionn, more new articles in various fields may be considered. For instance, The Epilepsy Detection by Different Modalities with the Use of AI-Assisted Models; Artificial Intelligence and Applications.
(6) Methodology: A brief description of the data annotation process should highlight the encountered problems and their solutions. The discussion of the YOLOv8 architecture should also cite the latest research and technological advances related to YOLOv8 to increase the persuasiveness of the article.
(7) Experiments and Results: The analysis of the experimental results should explore the reasons for variations in model performance, particularly in comparing different configurations or datasets.
(8) Conclusion: The main achievements and strengths of YOLOv8 in detecting broken pipes should be summarized clearly, emphasizing critical experimental parameters and dataset configurations for performance improvement. Additionally, discussing anticipated challenges in real-world deployments and plans for addressing them would enhance the conclusions.
(9) It would be beneficial to include the limitations regarding the limitations assessment, improving this section with greater rigor.
Author Response

(The authors gave the same response as above.)

Round 2
Reviewer 1 Report
Comments and Suggestions for Authors
The author has answered my question and made appropriate revisions to the paper, therefore it is recommended to accept the paper.
Reviewer 3 Report
Comments and Suggestions for Authors
accept